# Gender differences in affective reactivity to COVID-related daily stressors in mood and anxiety disorders: A six-month intensive longitudinal study

Dahlia Mukherjee[1,☉]*, Sun Ah Lee[2,3,☉], David M. Almeida[2,3], Erika F.H. Saunders[1]

**1** Department of Psychiatry and Behavioral Health, Penn State College of Medicine and Penn State Health, Hershey, Pennsylvania, United States of America, **2** Human Development and Family Studies, Penn State University, University Park, Pennsylvania, United States of America, **3** Center for Healthy Aging, Penn State University, University Park, Pennsylvania, United State of America

☉ Both authors made equal contributions and are first co-authors.
* dmukherjee@pennstatehealth.psu.edu

## Abstract

Gender differences in stress processes can contribute to the disparities in the prevalence of psychiatric disorders between women and men. However, gender differences in daily stress processes, particularly during the COVID-19 pandemic, among mental health clinical populations is yet to be explored. Our goal is to determine gender differences in daily stress processes related to the COVID-19 pandemic. This was an intensive longitudinal cohort study of outpatients with mood and anxiety disorders conducted from January 2021 to May 2023. Daily diary data consisting of self-reported daily positive and negative affect were collected across 8 consecutive days, each month, for a maximum period of 6 consecutive months. Multilevel modeling was implemented. Thirty-one women and 18 men (mean 41years [SD = 15]) completed 1,711 surveys over an average of 4.98 months. Multilevel models showed that women exhibited heightened affective reactivity to COVID-19 daily stressors. On days with an increased number of COVID-19 daily stressors, women experienced a decrease in positive affect (b = -0.08, SE = 0.03, $P$ = 0.007), but not men. At the person-level, a greater number of COVID-19 daily stressors was associated with higher negative affect (b = 0.28, SE = 0.08, $P$ < 0.001) only among women. In this novel, intensive longitudinal cohort study, women and men with mood and anxiety disorders did not differ in the frequency of COVID-19-related daily stressors. However, women were affectively more responsive to these stressors. Gender, therefore, may inform patterns of stress response in daily affective reactivity to COVID-19 stressors, which in turn, may inform stress-related treatment interventions.

**Data availability statement:** The name of the repository is Gender disparities in affective reactivity to COVID-19 stressor, and here is the updated OSF link: https://osf.io/2rsd3/overview?view_only=7f2c1e00f0d24f57909c-b18a5d2c5514.

**Funding:** This project was supported by funding from Penn State's Social Science Research Institute (SSRI) to DM and EFHS. Content is the responsibility of the authors and does not represent the views of the SSRI. The funders had no role in study design, data collection and analysis, decision to publish, or preparation of the manuscript.

**Competing interests:** The authors have declared that no competing interests exist.

## Introduction

Mood and anxiety disorders impose a substantial public health burden worldwide, contributing markedly to disability, mortality, and lost productivity [1], and have a bi-directional association with psychosocial and biological stress. Women suffer from mood and anxiety disorders at rates higher than men [2,3] and the discrepancy is especially marked during the reproductive years [4]. During the COVID-19 pandemic, the rates of anxiety and depression increased by 3–4 times, with a global increase by 25% of anxiety and depression according to the World Health Organization [5]. As highlighted recently by Mazure and Husky, one potential source for this increasing discrepancy is, compared to men, the difference in women's affective response to specific types and perceived occurrence of daily stressors [6].

*Daily* social and psychological stress is an understudied factor influencing the emergence and persistence of mental health disorders. While major life trauma has been studied extensively as a risk factor for psychiatric and mental illness [7,8], the impact of daily stressors on the development or maintenance of mood and anxiety symptoms in those vulnerable to major depressive disorders, bipolar disorders and anxiety disorder is less known [9]. Daily stressors are naturalistic, common minor hassles and events that arise from day-to-day living (e.g., work concerns and deadlines) or unexpected events that disrupt daily life (e.g., argument with a partner, malfunctioning computer) [10]. Daily stressors and affective response to daily stressors are a modifiable target via implementation of behavioral and mindfulness interventions for reducing risk and improving outcomes for individuals with mental health disorders [11,12]. Recent research suggests that chronic stress can be conceptualized and assessed through the patterns of acute, short-term stressors and their responses. In this framework, daily stressors, typically acute stressors with a clear onset and resolution (i.e., stressor-free days), offer a valuable window into the dynamic nature of stress as it unfolds in everyday life [13]. Daily stressors elicit immediate emotional and physical effects on the day they occur, such that they have more proximal consequences for both psychological and physiological function and are better predictors of long term physical and mental health outcomes than either major life events (e.g., death of a parent) or chronic stress (e.g., living in poverty) [14].

Importantly, the aggregate effects of daily stressors increase vulnerability to the development of future disease, likely via the cumulative influence of repeated spikes in stress system activation known as allostatic load [15]. Response to daily stressors is a modifiable factor that can be a treatment target. A recent study found that individuals with higher levels of daily stressor exposure, negative affect, and affective reactivity to stressors (i.e., an increase in negative affect on days with stressors) were at greater risk of developing depression over 10 years, suggesting daily stress processes as accessible intervention targets for reducing the long-term depression risk [11]. Poor metabolic and cardiovascular outcomes occur at higher rates in those with mental illnesses [16,17]. As daily stressors are increasingly linked to poor metabolic and cardiovascular outcomes, which in turn are linked to significant morbidity and mortality [18–20], it is worthwhile to understand the impact of response to daily stress in those with mental illness. The positive affect dynamics have been specifically

linked to physical health outcomes. For example, evidence suggests that PA variability is predictive of health outcomes through distinct pathways from those linked to average PA levels [21].

While the differential rates of mood and anxiety disorders prevalence between men and women and gender differences in daily stress response are known, it remains unclear whether affective response to stress will differ in men and women with mood and anxiety disorders. The existing evidence of gender differences in daily stress processes in a non-clinical population suggests that women are likely to experience greater stress occurrences and heightened affective response to daily stressors (e.g., arguments or avoiding arguments) [22–24]. In addition, such gender differences may reflect underlying gender socialization and role-expectation processes: women's greater involvement in unpaid labor and caregiving activities, and emotional/household management may both increase exposure to micro-stressors and amplify affective reactivity (for example, via higher emotional labor demands) [25,26]. Establishing potential sex or gender differences in affective response to daily stress in those with mood and anxiety disorders is important for several reasons. First, epidemiological data show that many mood and anxiety disorders have markedly higher prevalence in women than in men [27,28]. These prevalence differences raise the possibility that sex-specific psychophysiological or affective mechanisms contribute to risk for development of illness, and thus stress reactivity itself may show different patterns by sex in those with mood and anxiety disorders. Second, nonclinical and clinical studies have documented sex differences in stress physiology (e.g., cortisol responses, autonomic reactivity) and in emotional regulation more generally (e.g., women sometimes showing greater subjective negative affect or greater rumination under stress) [29,30]. If such differences extend into clinical populations, they might help explain heterogeneity in symptom expression, treatment response, or resilience. Finally, from a translational or precision-psychiatry perspective, sex differences in affective stress response might point to differential targets for intervention (e.g., gender-tailored stress regulation strategies).

Despite the theoretical importance, relatively few studies have been designed or sufficiently powered to test gender differences in affective stress reactivity in psychiatric samples, and the results have so far been mixed or inconclusive. Moreover, the COVID-19 pandemic created novel stressful experiences across multiple domains. The COVID-19 pandemic generated novel, multi-domain stressors—ranging from financial insecurity, social and relational disruptions, and job loss to illness risk and restrictions on mobility—which had pervasive impacts on mental health and daily well-being. Longitudinal evidence indicates that months characterized by greater COVID-related stressors were associated with heightened anxiety and depression and lower life satisfaction [31]. Similarly, data from the UK Household Longitudinal Study show significant within-person increases in psychological distress during pandemic waves, particularly among individuals experiencing caregiving and household burdens [32]. Daily-diary studies further demonstrate that on days with elevated pandemic worry, negative affect intensifies, and stress reactivity is magnified [33]. Together, these findings highlight the pandemic as a unique and multifaceted stress context that shapes daily emotional experiences and provides an essential backdrop for the current study's investigation of affective stress responses. Interpersonally, social distancing created an isolated atmosphere when the need for emotional and social support was high. Business restrictions led to layoffs and created financial and domestic stress. The uncertainty of when normalcy was going to return created anxiety and fear. Moreover, during extended periods of societal uncertainty — such as the prolonged phase of the COVID-19 pandemic from early 2021–2023 —gendered pathways may be accentuated, as changes in routine and differential distribution of household/work responsibilities lead to changes in interpersonal relationships. During the COVID-19 pandemic, the discrepancy in stress experienced by men and women may have been different, and gender differences may have been exacerbated. By monitoring daily stress occurrence, severity, and emotional response in a clinical sample of patients under treatment for psychiatric illness, we can identify how stress related to COVID-19 impacts affect and physical functioning, which in turn can lead to long term psychological and physiological health problems.

In this study, we investigated the role of daily stressors in affective response related to COVID-19 in women compared to men, in a cohort of treatment-seeking patients with mood and anxiety disorders in South Central Pennsylvania. To address this aim, we utilized daily diary methods where participants repeatedly report their emotions, behaviors, and experiences once

per day across consecutive days [34]. This method captures participants' experiences in real time, allowing for assessing within-person and between-person variability of everyday experiences while reducing recall bias and enhancing ecological validity [34]. We also examined gender differences in occurrences of COVID-related daily stressors and affective responses to these stressors during the COVID pandemic. We determined the feasibility of collecting daily affect and stressors information using a daily diary study paradigm for eight consecutive days, once a month, for six consecutive months.

## Methods

### Ethics statement

The study was approved by the Penn State institutional review board (IRB # 16212), and all participants provided informed consent. Data collection started 01/21 and ended 05/23. The study protocol was fully remote and approved by the Penn State IRB (#16212). Verbal consent was requested from participants. Potential participants were provided a description of the purpose of the study and study rationale, participant responsibilities, information to be collected, and risks and benefits of the study. Verbal consent approval was documented on the encrypted Penn State REDCap database.

### Data and study participants

Data were collected from two registries: the Penn State Psychiatry Clinical Assessment and Rating System (PCARES) Registry and the Penn State Mood Disorder Repository (MDR). PCARES is a quality registry of clinical data from patients seeking psychiatric and psychological care in a large teaching clinic in a mid-Atlantic health system [35–37]. The MDR includes participants who have participated in studies conducted in the Mood Disorder Lab at the Penn State Department of Psychiatry and Behavioral Health. Participants were recruited if they met the following criteria: (1) aged 18 years or older, (2) fluent in English, and (3) had reliable access to a device with an internet connection (e.g., laptop, desktop, smartphone, or tablet). Psychiatric diagnosis was not the part of the recruitment criteria as some patients may have been treated for psychiatric symptoms without a diagnosis. Of note, the federal COVID-19 public health emergency declaration ended on May 11, 2023 [38].

Once participants provided verbal consent, they completed baseline questionnaires on sociodemographic information, diagnosis of psychiatric conditions, health, depression (Patient Health Questionnaire-9; PHQ-9) [39], anxiety (General Anxiety Disorder-7; GAD-7) [40] and daily functioning (World Health Organization Disability Assessment Schedule 2.0; WHODAS 2.0) prior to beginning the daily diary protocol. Participants completed the daily diary surveys using Qualtrics software (Qualtrics, Provo, UT) which minimized participant burden. The daily diary survey can be completed on any device with internet connectivity (smartphone, tablet, laptop, desktop computer, etc.). Participants were given the option to select "prefer not to answer" for each question. Surveys started on the first Monday of every month and a reminder survey was sent on the preceding Friday. The daily emails with the survey link were sent at 6 pm every evening. A research team member sent the participant an email every day for the 8 consecutive days of the assessment. These emails contained information on how to access the daily diary survey. The participant was asked to complete the daily diary survey for 8 consecutive days, once per month, for 6 consecutive months. Participants were financially compensated for participating in the study for each month they completed the survey.

A total of 52 participants enrolled in the daily diary assessments, where daily diary surveys were conducted for eight consecutive days during each burst (month), with a maximum of six possible bursts. We ruled out two participants who had completed one day of daily surveys at each burst. The completion rate was 82.33%, and 50 participants completed 1,725 daily surveys over an average of 4.94 months.

### Daily diary measures

**COVID-19-related Daily Stressors.**  The daily diary survey included the Daily Inventory of Stressful Events (DISE) [41]. The DISE included asking about the types of daily stressors (e.g., argument with a friend, work/school deadlines,

malfunctioning computer), how the stressor made you feel (e.g., angry, nervous), and feelings of psychological distress (e.g., anxiety, irritability).

Based on the original DISE measures, items from various publicly accessible surveys on the effects of COVID-19 were modified for the use in daily administration and response [33,42]. Each study day, participants were asked whether they experienced any of the following COVID-19-related daily stressors: financial problems, inability to spend time with others, challenges at home, trouble obtaining supplies, distressing news reports, experience of physical symptoms of COVID-19, difficulty completing work or school requirements, and greater work or home responsibilities compared to before the COVID-19 pandemic. A dichotomous variable was created to represent the occurrence of any COVID-19-related stressor on each day (1 = yes, 0 = no), and the total number of COVID-19-related stressors was calculated for each day.

**Daily affect.** During daily surveys, participants rated the frequency of 15 positive emotions (i.e., in good spirits, cheerful, extremely happy, calm and peaceful, satisfied, full of life, close to others, like you belong, enthusiastic, attentive, proud, active, confident, motivated to do your daily activities, and enjoy doing your activities) and 15 negative emotions (restless or fidgety, nervous, worthless, so sad nothing could cheer you up, everything was an effort, hopeless, lonely, afraid, jittery, irritable, ashamed, upset, angry, frustrated, and tired) using a five-point scale (0 = none of the time, 1 = a little of the time, 2 = some of the time, 3 = most of the time, 4 = all the time) [43]. Daily positive and negative affect were calculated by averaging ratings of these items on each day. The within-person reliability was 0.93 for positive affect and 0.90 for negative affect, and the between-person reliability was 0.97 for positive affect and 0.98 for negative affect [44].

## Covariates

Age, education level, and the PHQ-9 scores were included as between-person covariates in the multilevel models. PHQ-9 was used to assess the severity of depressive symptoms [39], in which participants were asked to rate the frequency of nine clinical symptoms of depression over the past 2 weeks on a four-point scale. The average score of nine items was used for the analyses. The occurrence of non-COVID-19-related daily stressors was included as a within-person covariate. Daily stressors were measured by the DISE [41] in which participants reported whether each of six types of stressors had occurred in the past 24 hours: argument or disagreement, avoided an argument or disagreement, stressor at work/school, stressor at home, stressor that happened to a close friend or family member, and any other stressor. A dichotomous variable (1 = yes, 0 = no) was used for the analyses to represent the occurrence of any non-COVID-19-related daily stressors on a given day.

## Analytic plan

First, summary scores of demographic characteristics of participants, clinical measures of mental illness, and study completion rates were calculated. We ran logistic regression models to examine the effects of demographic and clinical measures on the study completion rates. Next, to examine the gender differences in COVID-19-related daily stressor occurrence, we first calculated the summary scores of these stressors by gender. Then, to explore within-person associations between COVID-19-related daily stressor and affect, we first calculated intraclass correlation (ICC) of daily affect. To adjust for the data structure where days are nested within individuals, we used multilevel modeling to test the main effects and interactive effects of COVID-19-related stressors and gender on daily affect. Multilevel modeling is an appropriate analytic tool for nested data structure and estimating within-person associations between variables and cross-level interaction effects of person-level variables on these relations [45]. We estimated the two-level random-intercept models where daily COVID-19-related stressors predicted on daily affect adjusting for demographic characteristics and PHQ-9 scores. We then included interaction terms between daily COVID-19-related stressors and gender to predict daily affect to determine the gender differences in affective reactivity to COVID-19-related stressors. We excluded one participant who identified themselves as nonbinary for the multilevel models due to the lack of power of this sample size. Three participants were removed for multilevel modeling due to missingness in covariates (i.e., age, education level, race, and PHQ-9). The

final analytic sample size was 49 participants with 1,711 daily observations. For the multilevel models, the sample size comprised 46 participants with 1,675 days for the positive affect model and 1,639 days for the negative affect model after. Day-level continuous variables were centered at person-mean, and person-level continuous variables were centered at grand mean, which allows the interpretation of the coefficients as deviations from person-specific mean and grand mean.

## Results

### Participants' characteristics

Participants' characteristics are presented in Table 1. The participant group (N = 49) had an average age of 41 years (SD = 15), and was primarily women (63%) and White (88%). The educational level of participants was generally high, with about 89% having completed some college or higher degrees. The mean PHQ-9 depression severity score was 9.75 (SD = 5.46). Women reported higher completion rates (OR = 1.17 [95% CI, 1.03 – 1.33], P = 0.020).

For clinical measures, the mean number of diagnosed mental disorders was 2.55 (SD = 1.65). The majority of the participants were diagnosed with major depressive disorder (MDD; 63%) and/or generalized anxiety disorder (GAD; 65%), followed by post-traumatic stress disorder (PTSD; 31%), bipolar disorder (29%), and panic disorder (20%).

**Table 1. Participants' Characteristics.**

| | Female (n = 31) | Male (n = 18) | Total (N = 49) |
|---|---|---|---|
| *Demographic and Completion Measures* | | | |
| Age, mean (SD) | 41.57 (14.87) | 40.61 (15.48) | 41.21 (14.95) |
| Race, *N* (%) | | | |
| White | 25 (80.65) | 18 (100.00) | 43 (87.76) |
| Black or African American | 2 (6.45) | | 2 (4.08) |
| Asian | 1 (3.23) | | 1 (2.04) |
| Others | 2 (3.23) | | 2 (4.08) |
| Education, *N* (%) | | | |
| Completed high school or GED | 4 (1.33) | 1 (5.56) | 5 (10.20) |
| Some college | 7 (23.33) | 5 (27.78) | 12 (24.49) |
| Associate degree | 1 (3.33) | 4 (22.22) | 5 (10.20) |
| Bachelor's degree | 10 (33.33) | 3 (16.67) | 13 (26.53) |
| Graduate or professional degree | 8 (26.67) | 5 (27.78) | 13 (26.53) |
| PHQ-9, mean (SD) | 9.33 (4.99) | 10.44 (6.25) | 9.75 (5.46) |
| Completion rates, mean (SD) | 87.38 (20.70) | 74.95 (25.44) | 82.81 (23.10) |
| *Clinical Diagnosis* | | | |
| Diagnostic status of mental disorders | | | |
| Total number of diagnosed disorders, mean (SD) | 2.52 (1.77) | 2.61 (1.46) | 2.55 (1.65) |
| Type of diagnosed disorders, *N* (%) | | | |
| MDD | 19 (61.29) | 12 (66.67) | 31 (63.27) |
| Bipolar | 8 (25.81) | 6 (33.33) | 14 (28.57) |
| GAD | 20 (64.52) | 12 (66.67) | 32 (65.31) |
| Social phobia | 3 (9.68) | 3 (16.67) | 6 (12.24) |
| Panic disorder | 5 (16.13) | 5 (27.78) | 10 (20.41) |
| OCD | 4 (12.90) | 2 (11.11) | 6 (12.24) |
| PTSD | 10 (32.36) | 5 (27.78) | 15 (30.61) |
| Others | 8 (25.81) | 2 (11.11) | 10 (20.41) |

*Note*. Completion rate = (total daily interview days)/ (8 * total bursts)

## The occurrences of COVID-19-related daily stressors

The summary statistics of COVID-19-related daily stressor occurrences are presented in Table 2. In general, participants experienced at least one COVID-19-related daily stressor on 25.83% of the total daily survey days (SD = 31.92%), with the mean total number of stressors per participant was 14.20 (SD = 33.96). The two most common COVID-19-related stressor were financial problems (27%) and hearing distressing news reports (14%). The average positive affect was 1.50 (SD = 0.67) and negative affect was 0.99 (SD = 0.69).

There was no statistically significant difference in the occurrences of COVID-19-related daily stressors between women and men. However, it is worth noting some of the results for descriptive purposes. Men reported a greater proportion of days with at least one COVID-19-related stressor (23% in women vs. 30% in men), but women reported a greater total number of COVID-19-related stressors (17.68 in women vs. 8.22 in men).

## Affective reactivity to COVID-19-related daily stressors

Between-person differences accounted for 52.8% of the total variance of daily positive affect and 65.1% of daily negative affect. The results from the multilevel models are presented in Table 3 and Fig 1. At within-person level, a greater number of COVID-19-related daily stressors were marginally associated with lower positive affect (Est. = -0.05, SE = 0.03, $P = 0.052$) and higher negative affect on the same day (Est. = 0.04, SE = 0.02, $P = 0.063$). At the between-person level, individuals who reported a greater number of COVID-19-related daily stressors were likely to report higher levels of negative affect (Est. = 0.23, SE = 0.08, $P = 0.003$).

There were significant interactions between COVID-19-related daily stressors and gender on daily positive and negative affect, which indicates that affective reactivity to COVID-19-related daily stressors differed by gender for both positive (Est. = -0.15, SE = 0.07, $P = 0.031$) and negative affect (Est. = 0.46, SE = 0.23, $P = 0.044$). Specifically, a greater number of *within-person* COVID-19-related daily stressors was associated with concurrent lower positive affect among women (simple slope Est. = -0.08, SE = 0.03, $P = 0.007$), but not men (Est. = 0.07, SE = 0.06, $P = 0.264$). For negative affect, a greater number of *between-person* COVID-19-related daily stressors were associated with higher negative affect among women (simple slope Est. = 0.28, SE = 0.08, $P < 0.001$), but not men (Est. = -0.18, SE = 0.22, $P = 0.405$).

**Table 2. Summary Statistics of COVID-19-related Daily Stressors.**

|  | Female (n = 31) | Male (n = 18) | Total (N = 49) | t | P value |
|---|---|---|---|---|---|
| % of COVID-related stressor days | 23.14 (29.96) | 30.45 (35.45) | 25.83 (31.92) | 0.77 | 0.446 |
| Total number of COVID-related stressors per person | 17.68 (41.61) | 8.22 (11.89) | 14.20 (33.96) | -0.94 | 0.353 |
| % of days with financial problems | 10.15 (26.34) | 13.06 (29.26) | 11.22 (27.18) | 0.36 | 0.722 |
| % of days with unable to spend time with others | 7.87 (20.66) | 0.94 (2.42) | 5.32 (16.74) | -1.42 | 0.165 |
| % of days with challenges at home/with others | 5.29 (16.69) | 0.58 (1.92) | 3.56 (13.44) | -1.19 | 0.241 |
| % of days with trouble obtaining supplies | 1.47 (5.28) | 0.12 (0.52) | 0.98 (4.24) | -1.08 | 0.288 |
| % of days with distressing news reports | 11.45 (21.60) | 19.54 (32.38) | 14.42 (26.05) | 1.05 | 0.299 |
| % of days with experiences of physical COVID-19 symptoms | 4.87 (7.52) | 5.98 (15.98) | 5.28 (11.22) | 0.33 | 0.744 |
| % of days with difficulty completing work | 2.10 (10.12) | 0.20 (0.84) | 1.40 (8.07) | -0.79 | 0.431 |
| % of days with difficulty completing school requirements | 0.53 (1.52) | 0.15 (0.65) | 0.39 (1.28) | -0.99 | 0.327 |
| % of days with greater work or home responsibilities | 4.53 (16.51) | 2.13 (7.88) | 2.58 (13.92) | -0.58 | 0.567 |
| Positive affect | 1.47 (0.62) | 1.56 (0.76) | 1.50 (0.67) | 0.48 | 0.634 |
| Negative affect | 0.93 (0.61) | 1.09 (0.83) | 0.99 (0.69) | 0.77 | 0.448 |

*Note.* T-tests were performed to test statistical difference in COVID-19-related daily stressors between men and women. T-tests were based on the assumption of equal variance which were determined by the Levene's test of equality of variance prior to t-tests. Degrees of freedom for t-tests were 47.

**Table 3. Results from the Multilevel Model Examining the Impacts of the Total Number of COVID-19-related Daily Stressors on Affect (N=46).**

| | Positive affect | | | | | | Negative affect | | | | | |
|---|---|---|---|---|---|---|---|---|---|---|---|---|
| | Est. | SE | P | Est. | SE | P | Est. | SE | p-value | Est. | SE | P |
| *Fixed effects* | | | | | | | | | | | | |
| Intercept | **1.64** | **0.38** | **<0.001** | **1.63** | **0.38** | **<0.001** | **1.17** | **0.25** | **<0.001** | **1.17** | **0.25** | **<0.001** |
| Age | -0.00 | 0.01 | 0.801 | -0.00 | 0.01 | 0.599 | 0.01 | 0.00 | 0.136 | 0.01 | 0.00 | 0.062 |
| Female | -0.05 | 0.22 | 0.813 | -0.05 | 0.22 | 0.831 | **-0.30** | **0.15** | **0.041** | **-0.30** | **0.14** | **0.034** |
| White | 0.11 | 0.34 | 0.752 | 0.12 | 0.34 | 0.720 | -0.31 | 0.23 | 0.170 | -0.32 | 0.22 | 0.143 |
| College graduate | -0.24 | 0.22 | 0.262 | -0.25 | 0.21 | 0.241 | 0.21 | 0.14 | 0.148 | 0.21 | 0.14 | 0.123 |
| PHQ score | **-0.04** | **0.02** | **0.041** | **-0.04** | **0.02** | **0.023** | **0.08** | **0.01** | **<0.001** | **0.08** | **0.01** | **<0.001** |
| Non-COVID19 str (WP) | **-0.21** | **0.04** | **<0.001** | **-0.22** | **0.04** | **<0.001** | **0.32** | **0.03** | **<0.001** | **0.33** | **0.03** | **<0.001** |
| COVID-19 str (WP) | -0.05 | 0.03 | 0.052 | 0.07 | 0.06 | 0.264 | 0.04 | 0.02 | 0.063 | -0.04 | 0.05 | 0.448 |
| COVID-19 str (BP) | -0.07 | 0.11 | 0.563 | 0.41 | 0.33 | 0.213 | **0.23** | **0.08** | **0.003** | -0.18 | 0.22 | 0.401 |
| COVID-19 str (WP) x Female | – | – | – | **-0.15** | **0.07** | **0.031** | – | – | – | 0.09 | 0.05 | 0.084 |
| COVID-19 str (BP) x Female | – | – | – | -0.53 | 0.35 | 0.122 | – | – | – | **0.46** | **0.23** | **0.044** |
| *Random effects* | | | | | | | | | | | | |
| Intercept | 0.44 | | | 0.42 | | | 0.19 | | | 0.18 | | |
| Residual | 0.35 | | | 0.35 | | | 0.21 | | | 0.21 | | |

*Note.* $N_{obs}$ = 1675 for positive affect models and $N_{obs}$ = 1639 for negative affect models. WP = within-person, BP = between-person

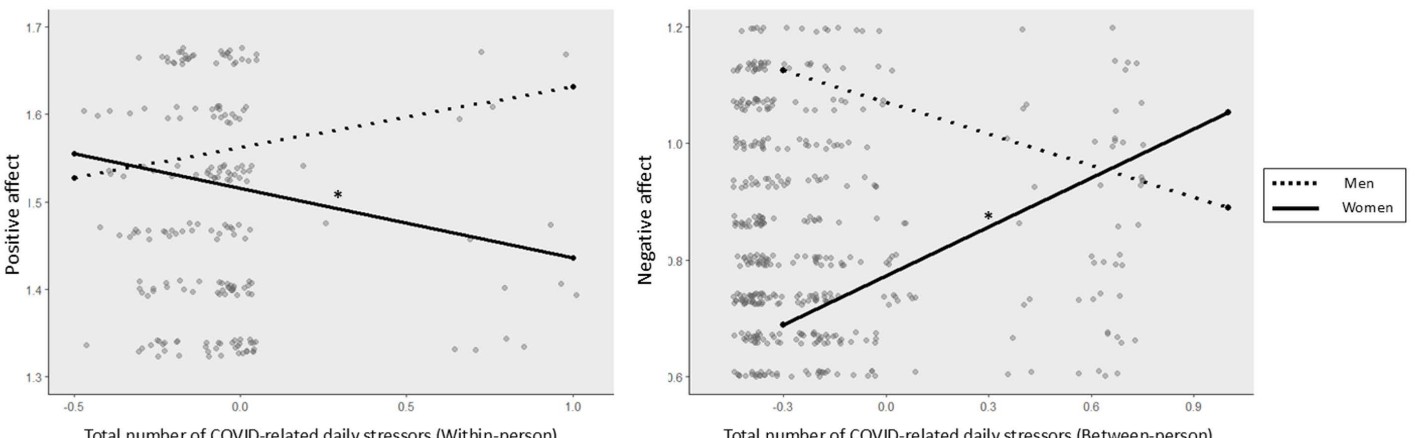

**Fig 1. Gender Differences in Affective Reactivity to COVID-19-related Daily Stressors.** *Note.* Gender moderated the associatoins between COVID-19-related daily Stressors and daily affect. A greater number of COVID-related daily stressors were associated with lower positive effect at the within-person level (left panel), and with higher negative affect at the between-person level only among women (right panel). The total number of COVID-related daily stressors was centered at the person-mean for positive affect and at the grand mean for negative affect. Simple slopes significant at p < .05 are denoted with an asterisk*.

## Discussion

The COVID-19 pandemic resulted in challenges and stressors that negatively impacted mental health and wellness and in the current study provided a unique opportunity to understand the affective response to daily stress during the pandemic by individuals seeking outpatient psychiatry treatment. This is the first study that has investigated the role of daily COVID stressors longitudinally across a six-month period in people with mood and anxiety disorders, designed to assess gender

differences [6]. Based on the completion rates, the results suggest that the method is feasible and acceptable to mental health clinical populations. The results from the current study demonstrated significant gender differences in the affective reactivity to COVID-19 stressors but indicated no gender difference in the occurrence of COVID-19-related stressors. Specifically, days with a greater number of COVID-19-related stressors were associated with lower positive affect and higher negative affect on those days, but only among women.

Women are at a higher risk for depression than men [46], and women are more vulnerable to stress and post-traumatic stress disorder than men [47]. In recent studies, the prevalence of anxiety, depression, and stress during COVID-19 pandemic was also found to be higher in women than in men [48,49]. This is not limited to COVID-19-related experiences, supported by findings that women generally report more frequent daily stressors compared to men [22]. Contrary to these findings, we did not find significant gender differences in the occurrence of COVID-19-related stressors in daily lives. This may be because – in retrospect – we know that our study was conducted at the end of the pandemic when the COVID-19 stress experiences were different from those in the early stages of the pandemic. By this stage, many COVID-specific contextual stressors had attenuated: infection rates and hospitalizations had declined in many regions, vaccination campaigns had reduced perceived threat, media coverage became less central to everyday life, and people had adapted to new routines (e.g., remote work, masking, distancing) and the acute uncertainty that characterized the early pandemic [32,50,51]. Early in the pandemic, women disproportionately experienced stressors linked to caregiving demands, household responsibilities, and employment disruptions [52,53]. As households, workplaces, and institutions adjusted over time, longitudinal evidence suggests some of these gendered differences in exposures and perceived burden decreased and adjusted over time, which may help explain why stressor occurrence was more similar across genders in our late-pandemic sample [54]. Interestingly, being unable to spend time with others appeared to be more frequent in women (8% vs. 1% of the total survey days) but was not statistically different between groups. Women have been found to be more reactive to interpersonal tensions [24], and we hypothesized that women would have more frequent relationship or interpersonal stressors, but our findings did not support this hypothesis [55]. Future studies are warranted to delineate which types of interpersonal stressors (e.g., relationship with family, friends, or colleagues at work) are more frequently experienced across different genders.

Despite no statistically significant gender differences in COVID-19-related stressor occurrences, we found significant gender differences in affective reactivity to these stressors. On the days when women encountered more COVID-related stressors than usual, they exhibited lower positive affect, suggesting heightened day-to-day affective reactivity to situational stress. Furthermore, women who experienced greater overall exposure to COVID-related stressors across the study period relative to those who did not, reported higher levels of negative affect, indicating more chronic stress-related distress. In contrast, these associations were not evident among men. These patterns together suggest that women may be both more emotionally reactive to daily stress and more vulnerable to the cumulative effects of stress exposure during the pandemic. This is consistent with previous studies which found heightened affective reactivity to general daily stressors [22–24]. We extended these findings by examining the effects of daily stressors specifically related to COVID-19. This is in line with other studies which found women exhibited increased affective vulnerability during COVID-19 pandemic compared to men. For example, one study found women appeared to be particularly sensitive to fear, worry, and threat compared to men [56]. Another study, examining the impact of the COVID-19 pandemic on academics, social isolation, and mental health amongst university students, determined a more pronounced negative effect on female students' academics, social isolation, stress and mental health compared to male students [57]. It is notable that there was no difference in average daily affect between women and men in the current sample. These findings, together, highlight gender disparities in affective reactivity to COVID-19-related stressors. Even though women generally exhibited similar affective states as men, changes in these states are triggered by COVID-19-related daily stressors only among women.

Importantly, the present findings also provide insights that extend beyond the COVID-19 pandemic. Periods of collective uncertainty— for example, whether due to natural disasters, economic crises, or sociopolitical instability—share features with pandemic-related disruption, including chronic unpredictability, altered daily routines, and shifts in work and family roles [58,59]. Understanding gendered patterns in stress reactivity under these conditions can therefore inform future mental health interventions aimed at bolstering resilience and adaptive coping in contexts of widespread uncertainty. For instance, interventions that target emotion regulation and social support among women, particularly those managing disproportionate caregiving or relational burdens, may buffer against the affective costs of acute and prolonged collective stress. Moreover, adaptive daily diary or ecological momentary assessment designs can be deployed in future crises to monitor fluctuations in affect and stress in real time, guiding personalized preventive and therapeutic responses [60].

## Limitations

The study was conducted at the end of the pandemic limiting findings to this time period. The findings may have been different depending on the type of COVID strain affecting individuals and/or restrictions imposed by the CDC. During this later phase of the pandemic, widespread vaccine availability, improved treatment options, and societal adjustment and adaptation likely reduced the intensity of emotional responses compared to earlier, more uncertain phases of the pandemic. The second limitation is limited racial and ethnic diversity, and a generally well-educated sample, which limits conclusions regarding the impact of stressors on different racial and ethnic groups. In addition, the one-time daily measure did not allow for within-day affective dynamics assessment and disentangling causality of changes in affect following stressor or vice versa. However, compared with more intensive approaches, such as ecological momentary assessments, which involve brief but more frequent data collection, the daily diary approach includes a single assessment each day, allowing participants to provide extended and detailed reflections of their daily experiences. Finally, we have not examined the impacts of contextual factors, particularly participant's social relationship (e.g., family composition, married/cohabit status). Future research should examine how these factors shape daily stress and affective experiences among individuals with mental illness.

## Conclusions

The findings demonstrate the importance of focusing on gender differences in stress response rather than the occurrence or frequency of daily stressors. Our findings highlight gender differences in patterns of stress response, which in turn, could potentially inform interventions to address the impacts of daily stress.

## Acknowledgments

The authors would like to thank Zachary Nitsch, Megan Brady and Natalie Marr for helping with data collection, Penn State SSRI for funding the project, and, the participants who provided their invaluable time.

## Author contributions

**Conceptualization:** Dahlia Mukherjee, David M Almeida, Erika FH Saunders.

**Data curation:** Dahlia Mukherjee, Erika FH Saunders.

**Formal analysis:** Sun Ah Lee.

**Funding acquisition:** Dahlia Mukherjee, David M Almeida, Erika FH Saunders.

**Methodology:** Dahlia Mukherjee, Erika FH Saunders.

**Project administration:** Dahlia Mukherjee, Erika FH Saunders.

**Supervision:** Dahlia Mukherjee.

**Writing – original draft:** Dahlia Mukherjee, Sun Ah Lee.

**Writing – review & editing:** Dahlia Mukherjee, Sun Ah Lee, David M Almeida, Erika FH Saunders.

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
