## [Decision Letter · Decision Letter 0]

17 Aug 2025

PMEN-D-25-00025

Gender linked to COVID-related daily stress processes in mood and anxiety disorders: A six-month intensive longitudinal study

PLOS Mental Health

Dear Dr. MUKHERJEE,

Thank you for submitting your manuscript to PLOS Mental Health and I am very sorry for the delay in reaching a decision. After careful consideration of the reviewer reports that we have now received, we feel that your paper has merit but does not yet fully meet PLOS Mental Health’s publication criteria as it currently stands. Therefore, we invite you to submit a revised version of the manuscript that addresses the points raised during the review process.

Please ensure that you address all of the comments raised, which you can find at the end of this email. 

We look forward to receiving your revised manuscript.

Kind regards,

Dr Karli Montague-Cardoso

Executive Editor

PLOS Mental Health

Journal Requirements:

1. In the ethics statement in the Methods, you have specified that verbal consent was obtained. Please provide additional details regarding how this consent was documented and witnessed, and state whether this was approved by the IRB

1. Please clarify all sources of funding (financial or material support) for your study. List the grants (with grant number) or organizations (with url) that supported your study, including funding received from your institution. 

2. State what role the funders took in the study. If the funders had no role in your study, please state: “The funders had no role in study design, data collection and analysis, decision to publish, or preparation of the manuscript.”

3. Please send a completed 'Competing Interests' statement, including any COIs declared by your co-authors. If you have no competing interests to declare, please state "The authors have declared that no competing interests exist". Otherwise please declare all competing interests beginning with the statement "I have read the journal's policy and the authors of this manuscript have the following competing interests:"

4. Please provide separate figure files in .tif or .eps format.

https://journals.plos.org/mentalhealth/s/figures 

https://journals.plos.org/mentalhealth/s/figures#loc-file-requirements

5. Please insert an Ethics Statement at the beginning of your Methods section, under a subheading 'Ethics Statement'. It must include:

1) The name(s) of the Institutional Review Board(s) or Ethics Committee(s)

2) The approval number(s), or a statement that approval was granted by the named board(s) 

3) (for human participants/donors) - A statement that formal consent was obtained (must state whether verbal/written) OR the reason consent was not obtained (e.g. anonymity). NOTE: If child participants, the statement must declare that formal consent was obtained from the parent/guardian.

Reviewers' comments:

Reviewer's Responses to Questions

**Comments to the Author**

1. Does this manuscript meet PLOS Mental Health’s publication criteria?

Reviewer #1: Yes

Reviewer #2: Yes

2. Has the statistical analysis been performed appropriately and rigorously?

Reviewer #1: Yes

Reviewer #2: Yes

3. Have the authors made all data underlying the findings in their manuscript fully available (please refer to the Data Availability Statement at the start of the manuscript PDF file)?

Reviewer #1: Yes

Reviewer #2: Yes

4. Is the manuscript presented in an intelligible fashion and written in standard English?

Reviewer #1: Yes

Reviewer #2: Yes

Reviewer #1: Thank you for including me in this review. The authors present a paper examining gender differences in daily social and psychological stress and corresponding reactivity to stressful stimuli related to COVID-19. They show increased affective responsiveness for women vs. men in responding to COVID-19 related stressors, even as the frequency of these stressors remained constant between them.

The authors’ work offers an important, understudied angle on a still-timely topic. Overall, the paper is quite clear and coherently presented. However, it would strongly benefit from some re-organization; additional detail and context at a number of points; and a more thorough visual representation of the results. My comments are listed below by quote and page number, with major recommendations followed by minor suggestions.

Major recommendations:

Page 3: “Women suffer from mood and anxiety disorders at rates higher than men…” While this opening introduces the topic of your paper, it feels like you are jumping into a specific, existing finding without any framing. I would recommend adding a sentence before this that generally frames why the work you’re doing is crucial, to grab the reader’s attention and contextualize this background further.

Page 3: “differential occurrence and affective responsivity to daily stressors” – I would give an example of a past finding to contextualize this. Additionally, it’s not clear what is meant by “differential occurrence…to daily stressors” – please make sure that this is coherent.

Page 3, bottom: “Daily stressors elicit immediate emotional and physical effects on the day they occur…” – you contrast this with the longer-term effects of chronic stress, but it is not clear to me where an acute stressor (e.g. receiving a troubling email) might fit in. Would you argue that these daily stressors are also acute stressors, given their shorter time course? If not, I would recommend briefly addressing the distinction between daily stressors and acute stressors as they contribute/relate to chronic stress.

Page 4: “and both are causes of significant morbidity and mortality” – are daily stressors causes of these, or linked to higher morbidity and mortality rates by way of other factors? I would speak briefly to what kind of relationship these factors have to morbidity and mortality, and ensure that you cite a source for this point.

Page 4: “it is unknown whether affective response to stress will differ in men or women with these disorders” – this seems like one of the main questions of your paper, yet it feels buried in this paragraph with little lead-in. I would add a little more information contextualizing why this question is important and what we do know about it so far.

Pages 4-5: “Moreover, the COVID-19 pandemic created novel stressful experiences across multiple domains.” – this appears to be the main focus of your paper. However, here it feels more like a minor point contextualizing your main question rather than the key circumstance under which the daily stressors you’re interested in arises. I would recommend leading a paragraph with this point, and perhaps citing a source (as well as for the sentences following it) discussing the impacts of the specific factors you’ve mentioned on people’s well-being. These may be self-evident but there’s also a lot of important research that can and should support your claims (e.g. Mukherjee & Pahan, 2021; Daly & Robinson, 2022, etc.).

Page 5: “the discrepancy in stress experienced by men and women may have been different, and gender differences may have been exacerbated.” – See above; there’s many great sources you could cite for these claims!

Page 5: “In this study, we investigated the role of daily stressors…” Up to this point, you haven’t discussed how you are going to investigate the impact of daily stressors. Briefly discuss and justify why a daily diary paradigm is the method you chose to examine your questions.

Page 5: Cite the PCARES and MDR databases.

Page 6: “Participants were free to skip any questions that they preferred not to answer.” In the results section, you should report the extent to which questions were skipped. If many questions were skipped by many participants, be sure to discuss potential impacts on your results in the discussion section.

Tables 1-2: While I appreciate your inclusion of the non-binary participant’s data, reporting information for n=1 in isolation could potentially enable participant identification. Be sure to verify that including this level of information for n=1 is permissible.

Figure 1: If possible, please include individual data points on this figure (e.g. a scatterplot) so that we can see the distribution of affect and daily stressors per participant

Page 11: Ensure that the stats you provide are not repetitive to those in your table – you can discuss the statistics at the end of the Results section with reference to the table.

Page 12: “The results show that the method is feasible and acceptable to participants.” – Is this something that participants reported in feedback to you, or is it inferred from completion rates? Please specify.

Page 12: “This may be because – in retrospect – we know that our study was conducted…” – in what way might the stress experiences have differed? For example, was COVID being reported less in the news? Were people around the participants getting it less often? Furthermore, how would this point explain the lack of observed gender differences in stressor occurrence? Would the progression of the pandemic and associated stressors have equalized the experience for women and men over time, for example?

Page 12: You discuss reasons for the expected gender differences in women quite thoroughly. However, in general, it would be important to further discuss/speculate why you may not have seen the expected gender differences in affective states as triggered by COVID-19 daily stressors.

Page 13, bottom: “The findings may have been different at depending on the type of COVID strain affecting individuals and/or restrictions imposed by the CDC.” – while this partly addresses my point above about time-dependent differences in the progression of the COVID-19 pandemic, I would again speculate further as to how each of the differences you mentioned might impact daily stressors (e.g. would perceived transmissibility, or decreased restrictions, be predicted to upweigh/downweigh responses to daily stressors respectively)?

Page 14: “However, the strength of the daily diary study allowed for more comprehensive information on stressors, response and other variables of interest.” Increased frequency/sampling of responses does not necessarily mean that these responses are more comprehensive. Furthermore, I don’t know what you mean by “comprehensive information on…response” (which should also be plural). I would tighten this part up and be more specific in your justification.

Page 14, end: I would recommend ending off with a more general/broadly applicable takeaway from your results, so that the reader knows the importance of your work to everyday life.

Minor suggestions:

Title: You may consider rewording the title to indicate the direction of the relationship found between gender and reactivity to daily stressors; at the very least, it might be helpful to mention reactivity instead of just “processes”.

Abstract: “particularly during the COVID-19 pandemic” – add a comma after this.

Abstract: Add “an” before “increase in negative affect”.

Abstract: Add a comma between “novel” and “intensive”, as well as commas around “in turn”.

Page 7: “Unable to spend time with others” – I might recommend the phrasing “inability to spend time with others” instead, unless this is a verbatim quote of this event.

Pages 7-8: I would add in “emotions” right before “and negative” as these are long lists and the reader could lose track of what the categories are.

Page 9: Add “the” before “positive affect model” and “negative affect model”.

Page 10: Should “symptom” be plural?

Page 12: add “the” to “stress during COVID-19 pandemic”.

Page 12, end: “gender” should be plural.

Reviewer #2: The current study utilizes daily diary data collected during the COVID-19 pandemic to examine gender differences in daily stress processes related to the COVID-19 pandemic, including exposure and affective reactivity to COVID-19-related daily stressors. Strengths of the study include the intensive longitudinal design, the quality of the data, the assessment of daily COVID-19-related stressors, and the data analysis.

In addition to these strengths, there are issues that warrant further consideration to further strengthen the study’s contribution.

Introduction:

The Introduction is well-structured and grounded in research in contemporary research on stress and health.

The Introduction would be further strengthened by elaborating more on potential explanations for the anticipated gender differences, including how exposure and reactivity to COVID-related stressors may be, in part, shaped by gender socialization, gendered expectations regarding division of labor, and gender differences in relationship monitoring and maintenance.

Further, given that the data was collected from January 2021 to May 2023, it would strengthen the contribution of the study to position the findings within a larger, less specific context, such as in times of uncertainty.

Methods and Results:

In addition to age, education level, and the PHQ-9, it would be helpful to explore, if available, additional covariates related to family structure, such as relationship status and number of children, which are likely to contribute to daily affect as well as COVID-19-related daily stressors.

The current study collected information on non-COVID-related stressors. It would be interesting to also consider gender differences in affective reactivity that consider how COVID-related stressors interact with non-COVID daily stressors. Considering how co-occurring daily stressors and COVID-related stressors contribute to affective well-being among people with mood and anxiety disorder would further strengthen the study’s contribution, particularly now that the significant daily stressors of the pandemic have ended by providing insights into gender differences in how everyday stressors interact with stressors associated with worry and uncertainty to shape affective responses within this under-studied population.

Discussion:

The authors acknowledge several study limitations, including the limited racial and ethnic diversity in the sample. Given that the sample was also generally well-educated, this limitation should also be acknowledged and considered.

Further, although the focus on COVID-19-related daily stressors is a novel contribution, given that we are several years past the most significant influences of the pandemic, it would strengthen the study to elaborate more on how the findings may inform interventions in response to other similar social contexts beyond the COVID-19 pandemic.

**Do you want your identity to be public for this peer review?** For information about this choice, including consent withdrawal, please see our Privacy Policy

Reviewer #1: **Yes: ** Brandon J Forys

Reviewer #2: No

---

## [Decision Letter · Decision Letter 1]

3 Dec 2025

Gender differences in affective reactivity to COVID-related daily stressors in mood and anxiety disorders: A six-month intensive longitudinal study

PMEN-D-25-00025R1

Dear Dr MUKHERJEE,

We are pleased to inform you that your manuscript 'Gender differences in affective reactivity to COVID-related daily stressors in mood and anxiety disorders: A six-month intensive longitudinal study' has been provisionally accepted for publication in PLOS Mental Health.

Best regards,

Karli Montague-Cardoso

Staff Editor

PLOS Mental Health

Reviewer Comments (if any, and for reference):

Reviewer's Responses to Questions

**Comments to the Author**

Reviewer #1: All comments have been addressed

publication criteria?

Reviewer #1: Yes

3. Has the statistical analysis been performed appropriately and rigorously?

Reviewer #1: Yes

4. Have the authors made all data underlying the findings in their manuscript fully available (please refer to the Data Availability Statement at the start of the manuscript PDF file)?

Reviewer #1: Yes

5. Is the manuscript presented in an intelligible fashion and written in standard English?

Reviewer #1: Yes

Reviewer #1: Thank you for making these thorough and thoughtful revisions. All of my feedback has been fully addressed. I only have some very minor proofreading recommendations:

Page 3:

Line 80: Omit the comma between “disorders” and “impose”

Line 87: Add a comma before “compared to men”

Page 6:

Line 157: “affective stress responses” should be plural

Page 7:

Line 182: I might use the term “associated with” instead of “lead to”, as causality may not be inferrable in this situation.

**Do you want your identity to be public for this peer review?** For information about this choice, including consent withdrawal, please see our Privacy Policy

Reviewer #1: **Yes: ** Brandon J Forys
